# Expanded HPV Genotyping by Single-Tube Nested-Multiplex PCR May Explain HPV-Related Disease Recurrence

**DOI:** 10.3390/microorganisms12112326

**Published:** 2024-11-15

**Authors:** Luiz Ricardo Goulart, Bruna França Matias Colombo, Mayara Ingrid Sousa Lima, Maria Socorro A. de Andrade, Juliana São Julião, Adriana Freitas Neves, Silma Regina Pereira

**Affiliations:** 1Institute of Biotechnology, Federal University of Uberlândia, Uberlândia 38400-902, MG, Brazilbrunafrancamatias@yahoo.com.br (B.F.M.C.); 2Department of Medical Microbiology and Immunology, University of California Davis, Davis, CA 95616, USA; 3Laboratory of Genetics and Molecular Biology, Department of Biology, Federal University of Maranhão, São Luís 65085-580, MA, Brazil; mayara.ingrid@ufma.br; 4Health Department of the State of Maranhão, São Luís 65000-00, MA, Brazil; socorro_carvalho@uol.com.br; 5BioGenetics Tecnologia Molecular Ltda., Uberlândia 38400-446, MG, Brazil; juliana@biogenetics.com.br; 6Institute of Biotechnology, Federal University of Catalão, Catalão 75705-220, GO, Brazil; adriana_freitas_neves@ufcat.edu.br

**Keywords:** HPV screening, cervical cancer, molecular diagnostics, molecular epidemiology

## Abstract

The role of the human papillomavirus (HPV) in the establishment of cervical cancer has driven studies to find more effective methods of viral detection so that early intervention strategies can be performed. However, the methods still have limitations, especially regarding detecting the different genotypes simultaneously. We have developed a high-throughput system using a single-tube nested-multiplex polymerase chain reaction (NMPCR) for the detection of 40 HPV genotypes using capillary electrophoresis. The NMPCR assay was compared to the Hybrid Capture 2 assay (HC2) with 40 women from the Northeast of Brazil (São Luis, MA), a high endemic region, where the HPV positivity was 75% and 37.5%, respectively. These results were validated by performing a molecular epidemiological study on 5223 Brazilian women undergoing gynecological examinations from 2009 to 2017, who presented with an HPV prevalence of 59%. Multiple infections were found in 62.5% and 58% of the patients from the endemic region and from the Brazilian women population, respectively, mostly presenting high-risk genotypes (90.5% and 60%, respectively). Considering cervical intraepithelial neoplasia and adenocarcinomas, the sensitivity and specificity were 97.5% and 100%, respectively. The NMPCR assay was also capable of identifying viral subtypes in cases of multiple infections, even with low viral loads (10^−6^ ng/µL of HPV DNA). The NMPCR test is a promising and robust tool for HPV diagnostics and a screening tool for prevention of cervical cancer.

## 1. Introduction

The human papillomavirus (HPV) is the most common sexually transmitted infection, and it has been identified as a definite human carcinogen in six types of cancer in the cervix [1], the penis [2], the vulva, the vagina, the anus, and in head and neck squamous cell carcinoma [3,4]. The World Health Organization reports that cervical cancer is the fourth largest female malignancy worldwide, affecting 662,301 women in 2022, with over 50% of these cases resulting in death [5]. In addition, the worldwide prevalence of HPV infection in women without cervical abnormalities is 11–12% [6]. Two effective prevention strategies for cervical cancer have been nominated—vaccination against the human papillomavirus (HPV) and cervical screening with primary HPV testing followed by treatment of precancerous lesions [7].

Despite the expectation of a substantial decrease in cervix cancer incidence after the implementation of the HPV vaccination programs, overall vaccination rates have been surprisingly low globally [1]. Several challenges must be addressed, particularly in developing countries, including reducing the financial cost of vaccines, incorporating adolescents into vaccination programs, and overcoming cultural barriers related to sexual education, among others [8]. It is worth mentioning that vaccination coverage is still restricted to four or nine HPV subtypes and, considering the wide and diverse distribution of the HPV genotypes in different populations, it is still mandatory for the periodic gynecological control of cervical cancer as well as the development of more efficient tools for HPV genotypes detection.

HPV DNA detection is considered the gold-standard technique, particularly when targeting the L1 structural capsid genome region. This region is conserved among different genotypes, with variations leading to the identification of more than 200 papillomavirus genotypes [9], of which approximately 100 are well characterized in humans. Among these, nearly 50 different types infect the genital tract [10], with high-risk genotypes including types 16, 18, 31, 33, 35, 39, 45, 51, 52, 56, 58, 59, 66, 67, 68, 73, and 82. Other types, such as 13, 26, 30, 32, 34, 40, 53, 54, 55, 61, 62, 64, 67, 69, 70, 71, 72, 74, 81, 83, 84, CP141, CP8304, CP4173, and CP8061, present intermediate risk. In contrast, types 6, 11, 40, 42, 43, 44, 57, and CP6108 are classified as low risk and are primarily associated with anogenital warts [11,12,13,14,15].

A strong correlation between viral integration into the host DNA and invasive carcinoma has led to the increasing use of DNA detection technology as a diagnostic tool to enhance the sensitivity and accuracy of HPV DNA detection [16,17,18,19,20,21,22]. Currently, the Hybrid Capture 2 assay (HC2, Qiagen, Redwood City, CA, USA) is the most commonly used test for HPV diagnosis, but it has some disadvantages, such as the inability to perform genotyping, lack of specimen adequacy control, the requirement for a larger amount of biological material, and the potential for cross-hybridization with low-risk genotypes [23]. Real-time PCR is highly sensitive, and although it can distinguish among various genotypes, the protocol requires parallel reactions with multiple fluorescent probes. Alternatively, other methods allow simultaneous amplification of broad-range DNA regions by using general primers [16] combined with specific oligonucleotides in a technique known as nested-multiplex PCR (NMPCR) [24]. Although NMPCR has been employed in some studies for HPV detection [25,26], these efforts have been constrained to a limited range of viral types. As a result, a diagnostic assay capable of comprehensively identifying the predominant viral types that infect the genital tract has not yet been developed.

Screening for cervical cancer using DNA and cytopathology has been reviewed, and the US Preventive Services Task Force recommends screening every 3 years with cervical cytology alone, every 5 years with high-risk HPV (hrHPV) testing alone, or every 5 years with hrHPV testing in combination with cytology (co-testing) for women aged 30 to 65 years [27]. However, we hypothesize that the lack of sensitive and comprehensive detection of most mucosal HPV genotypes, coupled with multitype infections, may lead to shifts in the frequency of virus genotypes under specific vaccination programs in the near future. This could reduce the burden of HPV types 16, 18, 6, and 11 while increasing the likelihood of the emergence of other virus subtypes with lower prevalence in the population.

Therefore, this investigation aims to report a new high-throughput assay approach based on NMPCR technology followed by automatic capillary electrophoresis (CE) to simultaneously detect 40 genotypes of HPV. This methodology allows for the detection of multiple infections, demonstrating its performance in epidemiological studies.

To assess the effectiveness of NMPCR-CE for HPV-type detection, we evaluated its sensitivity and specificity in comparison to both conventional cytopathological analysis and the Hybrid Capture 2 assay (HC2). An extensive epidemiological study was conducted in a population of Brazilian women undergoing routine gynecological follow-up. The results and their implications for diagnosis and clinical follow-up are discussed.

## 2. Materials and Methods

The present study was designed to report a new molecular assay based on a one-tube nested multiplex PCR (NMPCR) that simultaneously detects 40 HPV genotypes, which was compared to the Hybrid Capture 2 and cytopathological exams. Moreover, we performed a molecular epidemiological study of the Brazilian female population to validate the new HPV NMPCR assay (Figure 1).

### 2.1. Patient Accrual and Sample Collection

The design of the HPV screening study is depicted in Figure 2. In the first phase, we evaluated the performance of the novel assay in comparison to the commercially available Hybrid Capture 2 (HC2) assay. For this purpose, 40 patients were consecutively recruited during routine gynecological examinations, including cytopathological tests, in São Luís, Maranhão State, Brazil, a region with high HPV endemicity. The study was conducted in accordance with the Declaration of Helsinki, it was approved by the Research Ethics Committee of the Federal University of Maranhão (3060/2008-47), and all participants provided informed consent, with their identities remaining anonymous. The analyses followed a double-blind protocol, with samples independently tested using NMPCR and HC2 in two separate laboratories: BioGenetics (Uberlândia, MG, Brazil) and Digene (São Paulo, SP, Brazil). The mean age of the patients was 37.5 years old (ranging from 18 to 81 years of age). Clinical data were obtained from routine gynecological evaluation by Papanicolaou (Pap smears) testing (analysis of medical records) and the collection of cervical samples for detection of HPV genomic DNA. 

In the second phase, a comprehensive epidemiological study was conducted across the Brazilian female population for HPV screening, encompassing all the states of Brazil. This large-scale study included 5233 cervical swab samples collected during routine gynecological exams from various clinics across Brazil between 2009 and 2017. The analyses were performed at the BioGenetics Diagnósticos Laboratory (Uberlândia, Minas Gerais, Brazil) using NMPCR for the detection and genotyping of 40 HPV subtypes. Genital specimens were collected using endocervical and ectocervical swabs placed in sterile tubes containing 1 mL of lysis buffer (10 mM Tris-HCl, 2 mM EDTA, and 400 mM NaCl), and maintained at room temperature.

Among this cohort, 170 patients from Uberlândia, MG (Mid-Central Brazil) aged 25 years and older were specifically monitored for cytological abnormalities, including cervical intraepithelial neoplasia (CIN). All participants provided informed consent. The state of Minas Gerais serves as a representative sample of the Brazilian population due to its rapid growth and influx of migrants from across the country. This study was also conducted in accordance with the Declaration of Helsinki, and it was approved by the Research Ethics Committee of the Federal University of Uberlândia (CEP/UFU 013/1999).

### 2.2. DNA Extraction from Cervical Samples

In the first phase of the study (São Luis Cohort, N = 40 patients), genital samples were placed in a 500 µL lysis buffer (400 mM NaCl, 50 mM EDTA, and 25 mM Tris-HCl, pH 8.0), vigorously agitated, and then transferred to a new microtube for DNA extraction. Proteinase K (10 µL) and 10% of sodium dodecyl sulfate (SDS) (10 µL) were added for protein degradation. Microtubes were incubated in a water bath at 60 °C for a minimum of 2 h. After incubation, a saturated NaCl buffer was added to 1/3 of the sample volume, and tubes were further incubated for 10 min at −20 °C. Samples were then centrifuged at 11,000 rpm for 15 min to precipitate cell debris. The liquid supernatant was transferred into a new tube in which a 2X volume of absolute ethanol was added for DNA precipitation that was centrifuged at 11,000 rpm for 15 min. The precipitated DNA was washed with 70% ethanol, resuspended in about 20 µL deionized water, and stored at 4 °C for subsequent PCR. In the second phase, genomic DNA extraction for the cohort involved in the epidemiological study was also manually conducted between 2009 and 2017, using cell samples collected via cervical swabs and applying the salting-out technique. [28].

### 2.3. Outer Multiplex PCR Conditions

For the first round of amplification (the outer PCR reaction), one consensus MY09 primer, as previously reported [12], was used. Additionally, nine newly designed degenerate primers (four MY09BG and five MY11BG) were introduced as a modified version of the original MY09/11 primer system [12] to cover the sequences of forty HPV genotypes. The constitutive β-globin (BG; GenBank Entrez U01317) gene was employed as an internal positive control for the reaction and to assess the efficiency of DNA extraction (Table 1).

The PCR mixture was performed in a final volume of 30 µL, containing 7 µL of DNA (5 ng/µL), 50 mM of KCl, 10 mM of Tris-HCl (pH 8.3), 1.5 mM of MgCl_2_, 120 µM of final concentration of deoxynucleoside triphosphate (dNTP), 1.5 U of AmpliTaq Gold DNA Polymerase (Thermo Fisher Scientific, Waltham, MA, USA), 0.675 pmol of each MY09/MY11 primer, and 2 pmol of each BG1 and BG2 primers. The reaction was incubated for 40 cycles at 94 °C for 1 min, 50 °C for 1 min, 72 °C for 2 min, and a final extension cycle at 72 °C for 5 min. 

### 2.4. Nested-Multiplex PCR (NMPCR) Conditions

In the second reaction, 40 specific primers labeled with fluorophores were arranged in a cocktail multiplex PCR. They were designed to identify the high-risk (16, 18, 26, 30, 31, 33, 34, 35, 39, 45, 51, 52, 53, 56, 58, 59, 66, 67, 68, 69, 70, 73, and 82), medium-risk (54, 55, 61, 62, 64, 71, 72, 74, 81, 83, and 84), and low-risk genotypes (06, 11, 42, 43, 44, and 57). The BG2 with BG3 primers for the constitutive β-globin gene were used. All primer sequences used are presented in Table 2.

The second PCR amplification had a final volume of 15 µL: 50 mM of KCl, 10 mM of Tris-HCl (pH 8.3), 1.5 mM of MgCl_2_ enhancer (1.5X), 60 µM of dNTPs, 1.0 U of AmpliTaq Gold DNA Polymerase (Thermo Fisher Scientific), 0.5 pmol of each HPV specific primer, and 0.3 pmol of control primers (BG2 and BG3). Two microliters of the MY11/09 PCR products were used as template for nested PCR amplification. The reaction was incubated for 36 cycles at 95 °C for 2 min, 56 °C for 1 min, 72 °C for 40 s, and a final extension cycle at 72 °C for 5 min. Additionally, we performed serial dilutions to determine the minimum concentration capable of detecting the viral subtypes, ranging from 10^−3^ to 10^−6^ µg/µL, with successful detection even at concentrations as low as 10^−6^ µg/µL.

### 2.5. Capillary Electrophoresis (CE)

The amplified product from NMPCR was diluted 200X, and 1 µL was submitted to capillary electrophoresis (MegaBase 1000, Amersham Pharmacia Biotech, Piscataway, NJ, USA) using the internal standard molecular weight ETRox550 (GE Healthcare, Chicago, IL, USA) for samples analysis (Figure 1). For the electropherogram analysis, we have built a new peak filter with the software Fragment Profiler 1.2 (GE HeathCare) with three fluorescein-labeled probes: FAM (6-carboxyfluorescein), HEX (5′-hexachlorofluorescein phosphoramidite), or TAMRA (6-carboxytetramethylrhodamine) (Table 2). The results were classified as positive in the presence of amplification for one (single infection) or more HPV viral types (multiple infections), generating 74–338 bp fragments, with the presence of the internal control amplification (β-globin gene-BG, 366 bp). The results were considered negative when only the β-globin (BG) gene was amplified and indeterminate when no amplification was observed for either HPV or the BG control.

### 2.6. Hybrid Capture 2 Assay (CH2)

HPV DNA detection was performed by the commercially available Hybrid Capture 2 assay (HC2, Qiagen) at Digene (São Paulo, Brazil). The HC2 assay is a nucleic acid hybridization assay with signal amplification that utilizes microplate chemiluminescent detection. Samples collections were performed in an appropriate tube provided in the HC2 kit. All scrapes were analyzed for the presence of low-risk HPV types 6, 11, 42, 43, and 44 and for the high-risk HPV types 16, 18, 31, 33, 35, 39, 45, 51, 52, 56, 58, 59, and 68. The HC2 positive results were categorized as HPV-A (Low-Risk), HPV-B (High-Risk), or both HPV-A/-B genotypes.

### 2.7. Statistical Analysis

Statistical analyses, tables, and graphs were produced using BioEstat (version 5.3) and GraphPad Prism (version 5) software. The detection of high-grade CIN2+ and CIN3+ lesions were assessed using sensitivity and specificity parameters. The detection of HPV in all CIN+ lesions were also determined in both São Luís (Maranhão) and Uberlândia (Minas Gerais), Brazil.

## 3. Results

### 3.1. The Nested-Multiplex PCR and Diagnostic Parameters 

The length of the amplicons generated by the first amplification with MY09/11 degenerated primers was 450-bp (Figure 3A). The β-globin gene detection rate was ≥90%, validating the DNA extraction method for HPV detection (Figure 3B). The HPV detection and genotyping by NMPCR were confirmed in agarose gel electrophoresis by selecting amplified samples with single infections ordered by size (Figure 3C) and also in multiple infections (Figure 3D). DNA amplicons from single and multiple infections from 40 randomly selected women representing all genotypes were excised from the gel, purified, and their sequences were confirmed by DNA sequencing. Additionally, the single-tube NMPCR was able to detect differences in viral load in double or multiple infections (Figure 3E) when serial dilutions were applied to samples prior to amplification, indicating that a decreased viral load of samples with multiple infections may result in detection failure, either by using conventional PCR or by the HC2 assay.

HPV detection using NMPCR in patients from São Luís-MA (N = 40) was approximately twice as high as that obtained through HC2 and cytopathology assays (Table 3), with positivity rates of 75%, 30%, and 40%, respectively. Among the infected patients, at least one high-risk HPV genotype was identified in 86.7% (26/30) of cases. Additionally, 83.3% (25/30) of the women had multiple infections. Remarkably, 71.4% (20/28) of the patients who tested negative by the HC2 assay were found to be positive by NMPCR, with 85.0% of these harboring high-risk HPV and 47.0% already exhibiting cytological lesions. Conversely, among the 10 patients who tested negative by NMPCR, only 2 (20%) were positive by HC2, and, notably, no cytological lesions were observed in these patients. Detailed comparisons of the three assays for each patient are provided in Appendix A.

### 3.2. Epidemiological Data

The Brazilian female population presented an HPV prevalence of 59.0%, among whom 58.0% had multiple infections and, among them, 60% harbored hrHPV genotypes. The frequency of HPV genotypes are presented in Table 4, which shows higher prevalence of single infections by HPV-6 (15.1%), followed by hrHPV-53 (7.1%), 31 (5.2%), and 16 (4.2%); however, among those with multiple infections, in addition to HPV-6 being the most prevalent (27.5%), HPV types 52, 54, 53, 42, 61, 51, and 16 were present in 13.4% to 18.3% of the cases, suggesting that the majority of cases have multiple infections, including a mixture of low- and high-risk genotypes. Appendix A shows the HPV DNA prevalence in female genitals from other studies.

We emphasize that 170 Papanicolaou samples from Uberlândia (Minas Gerais, Brazil) were clinically monitored for CIN+ lesions: 54.1% did not show cellular abnormalities, 21.2% were CIN1+, 10% were CIN2+, 13.5% were CIN3+, and 1.2% were classified as having adenocarcinoma. In addition, among 99 NMPCR negative samples, 15% showed cell abnormalities that were probably due to causes other than HPV infection. Considering CIN2+ and CIN3+ lesions only (40 out of 170), both the sensitivity and specificity of the NMPCR assay were 100%. Considering all CIN+ lesions and adenocarcinoma from both cities (São Luís-MA and Uberlandia-MG), the sensitivity and specificity were 97.5% and 100%, respectively. It is important to note that among three negative cases, two with CIN1+ lesions and one with adenocarcinoma, were also negative for HC2. Interestingly, among 80 specimens with CIN+ lesions from Uberlândia, we have also shown that as the disease progressed to severe lesions, single infections tended to predominate with hrHPV genotypes, culminating in adenocarcinoma with 1 virus or none at all (Figure 4), probably due to important genetic changes that may have led to the loss of cell architecture and physiological functions. Our study has also revealed important epidemiological data regarding the infection related to a woman’s age, i.e., there was a higher incidence of HPV infections in women between 19 and 30 years old and between 56 and 85 years old, mainly those with multiple infections.

## 4. Discussion

This study reports a novel high-throughput nested-multiplex PCR assay that uses a highly sensitive and specific broad-spectrum HPV DNA amplification on a single tube that allows the simultaneous detection of 40 common genital HPV genotypes. The results present significant epidemiological data that may change public health programs and may become an important molecular tool to monitor populations before and after HPV vaccination.

The viral genome contains highly conserved regions, for which several consensus PCR primer sets have been designed aiming for general HPV DNA detection, such as MY09/MY11 [16], GP5+/6+ [20,29], and SPF1/2 [30], resulting in distinct amplification of HPV DNA. However, the MY09/MY11 system has shown not only greater sensitivity than the other systems but also allows the identification of specific HPV genotypes. 

The nested-multiplex PCR assay (NMPCR) uses 1 of the original MY09/11 primer sets for the first PCR reaction with 9 additional degenerated primers to amplify all 40 genital HPV genotypes. The nested (second-round) specific amplification reaction was performed simultaneously in a single tube with 40 primer pairs with great efficiency and sensitivity, and it was designed to separate each virus genotype based on different amplicon sizes. Amplicons were separated into three sets of fluorophores to facilitate detection (Figure 1; Table 1 and Table 2). It is important to emphasize that the degeneration of primers in the first round of amplification was necessary to cover all the genomic variations within and among genotypes.

The technology proved to be highly precise and specific and capable of being performed even with small sample quantities. Although nested reactions may present the possibility of cross-contamination due to excess handling, we have observed no contamination. Our handling process of the nested reactions used as a general rule the individual manipulation of each sample at a time to prevent first-reaction spray and carry over. In order to demonstrate that reactions were not contaminated, random samples with multiple infections were individually tested with specific primers for those viruses previously detected by automatic capillary electrophoresis. All results were confirmed by agarose gel electrophoresis in separate reactions for each virus.

Our study involving 5223 patients consecutively recruited in routine gynecological examinations showed an HPV prevalence of 59.0%. In 2020, Wendland et al. [31] conducted an epidemiological study of HPV in Brazil with 5268 women and 1120 men who were sexually active and unvaccinated, aged between 16 and 25 years. The authors used a linear array (Roche) as a genotyping method, which detects 37 types of HPV, although only 13 high-risk genotypes, demonstrating a prevalence of HPV infection in 54.6% of the female population, corroborating our data. However, HPV prevalence varies across populations and is shaped by differences in diagnostic methods, sampling sites, population behaviors, cultural practices, and socioeconomic development indices.

Our new assay identified viral genotypes whose prevalence had not been investigated before, and, most importantly, it showed an extensive epidemiological profile of multiple and single infections not previously explored. A study carried out in the Northeast of Brazil with 250 women found 48% with HPV infections, with the most common type being HPV16 [32]. A large epidemiological study performed in Rio de Janeiro with 5833 women [33] detected the presence of HPV through the HC2 system with a positivity of 44.9%, with 25.5% related to high-risk genotypes. None of those studies were carried out by consecutive sampling without bias, which means that, in these cases, most of the women could be considered at risk or be suspected of having an HPV infection. It is important to note that clinicians at laboratories used for HPV detection or cancer screening usually refer women in Brazil when they present altered cytology or clinical signs. Comparisons across studies are presented in Appendix A [31,32,33,34,35,36,37,38,39,40,41,42].

Our large Brazilian epidemiological study revealed a predominance of multiple infections characterized by a mixture of low- and high-risk viruses at varying titers. As expected, an increased detection of HPV genotypes correlates with a higher prevalence. This suggests that undetected genotypes may become more significant over time, as type-specific vaccines might favor viral selection and amplification. In addition, detecting multiple infections is crucial, as patients with co-infections are at a higher risk of treatment failure [43]. In our study, the presence of low-titer, high-risk viruses alongside high-titer, low-risk viruses may have impacted the HC2 assay results. This is likely due to the HC2 assay’s use of a probe pool representing both low- and high-risk genotypes, where competition between probes could affect the accuracy of the detection. Therefore, the method presented in this study offers a robust approach for epidemiological surveillance, facilitating precise monitoring and timely interventions that promote better clinical outcomes.

Comparisons among NMPCR, HC2, and cytopathology assays also demonstrated false negative results for both HC2 and cytopathology, with a 50% and 40% rate of misdiagnosis, respectively, which may significantly impact public health programs and disease management, consequently hampering the measures used to prevent and control HPV infection by allowing the continuity of the transmission chain. Note that the discrepancy between genotyping and cytopathology can be attributed to frequent false-negative smears [35], which may be due to inadequate sampling or misinterpretation of the results [44]. On the other hand, false-positive Pap smears may be related to other cervical diseases unrelated to HPV [45,46,47]. Furthermore, women who have reached post-menopause or with more than 70 years of age are likely to present genital atrophy that can lead to increased inaccurate results [48,49]. Additionally, the HC2 assay not only has a limitation in identifying genotypes but also exhibits cross-reactivity between low-risk and high-risk groups.

Recent molecular approaches have emerged for clinical practice with high sensitivity and specificity, which are characterized by HPV detection and genotyping based on signal amplification methods (hybridization techniques in liquid phase) or target amplification methods (PCR amplification) with coverage varying from 15 to 34 HPV genotypes [26,50]. Our novel NMPCR tool coupled with capillary electrophoresis covers 40 genotypes, and in addition to its utility in HPV detection and genotyping, it is also able to determine the genotypic dominance in multiple infections, demonstrating that co-infections with prevalent low-risk genotypes may mask the detection of high-risk ones, especially when conventional PCR or hybridization assays are used. Another important finding in this study was the decreased number of multiple infections during disease progression, resulting in the occurrence of a single high-risk virus or complete viral loss in the adenocarcinoma stage. It is important to emphasize that during cancer development, modulation of transcription factors may inhibit viral replication, leading to a reduced number of viral copies in the cell, which may become undetectable by molecular techniques [51]. 

This study has certain limitations. Not all HPV-positive women were referred for colposcopy due to the wide geographic distribution of the clinics involved in sample collection across the country. Comprehensive cervical sampling followed by colposcopy was conducted in two cities, São Luís (MA) and Uberlândia (MG); thus, the specificity of the NMPCR assay may have been affected by this limited scope of follow-up.

## 5. Conclusions

The novel nested-multiplex PCR method described in this study represents a highly promising and robust tool for HPV diagnostics, with applications extending beyond population screening to the clinical follow-up of patients. Its capability to simultaneously detect, genotype, and assess viral dominance within a single tube has yielded valuable new epidemiological data. This new method may significantly impact clinical management and treatment strategies as well as influence cervical cancer statistics by facilitating the detection of subclinical infections and enabling the effective monitoring of populations, particularly in endemic regions.

## 6. Patents

Patent applications: BR1020170046150, filed in 8 March 2017 and WO/2018/161140, filed on 8 March 2018.

## Figures and Tables

**Figure 1 microorganisms-12-02326-f001:**
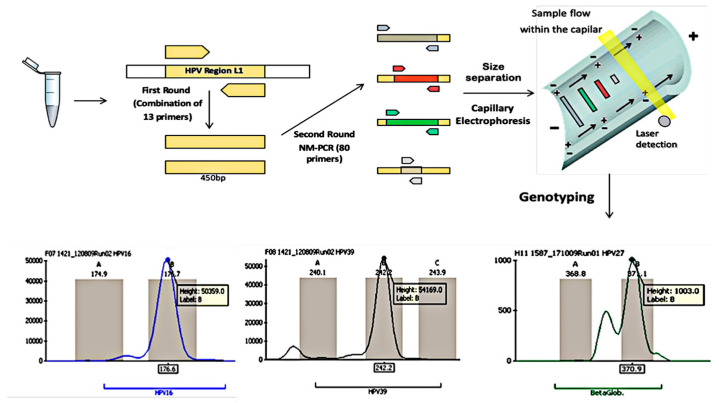
Schematic representation of the nested-multiplex polymerase chain reaction (NMPCR) for HPV detection and genotyping. It demonstrates a sample with double infection with HPV16 and HPV39, using β-globin as a reaction control. Although the peak intensities of both HPV genotypes are similar in this case, they may differ based on differential virus titers, indicating virus dominance for those with high intensity peaks.

**Figure 2 microorganisms-12-02326-f002:**
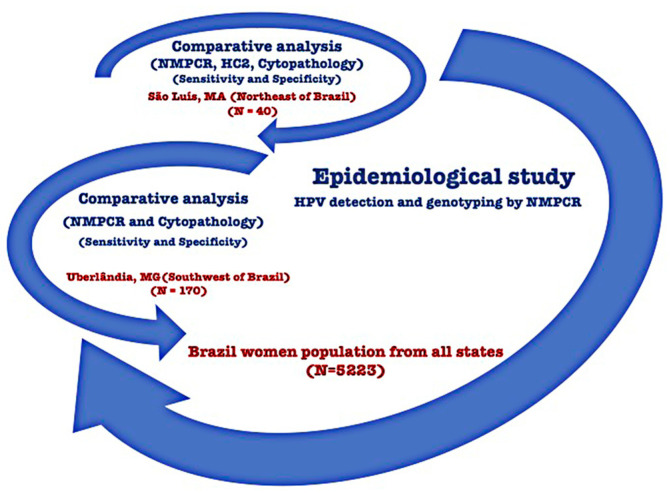
Study design applied to the Brazilian population (N = 5273).

**Figure 3 microorganisms-12-02326-f003:**
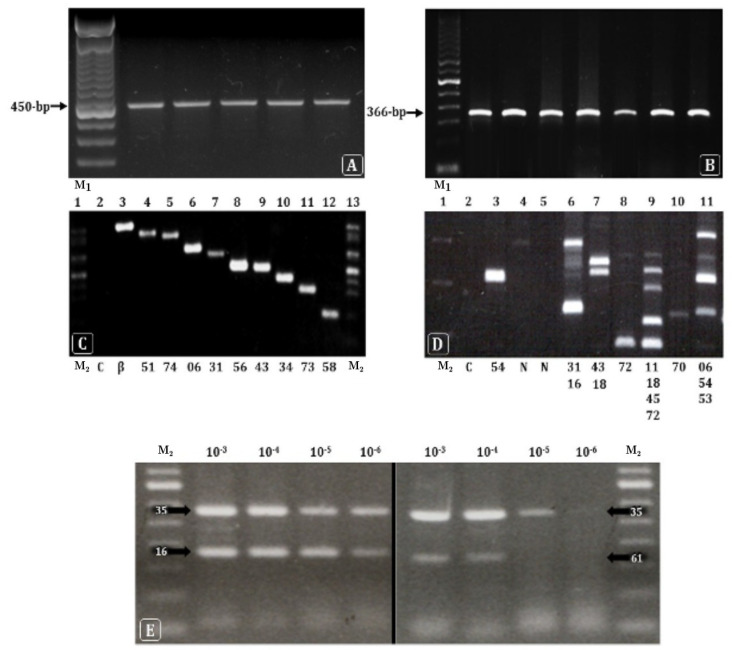
HPV genotypes of random samples and controls amplified by NMPCR as detected in agarose gel electrophoresis. (**A**): Example of first-round amplification (outer-PCR reaction) using 10 degenerated primers, generating 450-bp amplicons. (**B**): Examples of first-round amplification of the internal control of PCR reaction (β-globin gene), generating fragments of 366-bp. (**C**): Amplification of 11 single HPV types ordered by size. (**D**): Detection of single and multiple HPV infections in cervical samples. (**E**): Viral dominance detected by sample dilution of double HPV infections amplified by NMPCR; the left panel shows double infection with HPV16/35 with similar titers, and the right panel shows double infection with HPV35/61 with significant differences in titers (10^−3^ µg/µL to 10^−6^ µg/µL), with dominance of HPV35. Viral types are represented below in (**C**,**D**). Abbreviations: C, negative control; β-globin gene; M_1_, 100 bp molecular marker; M_2_, 50 bp molecular marker.

**Figure 4 microorganisms-12-02326-f004:**
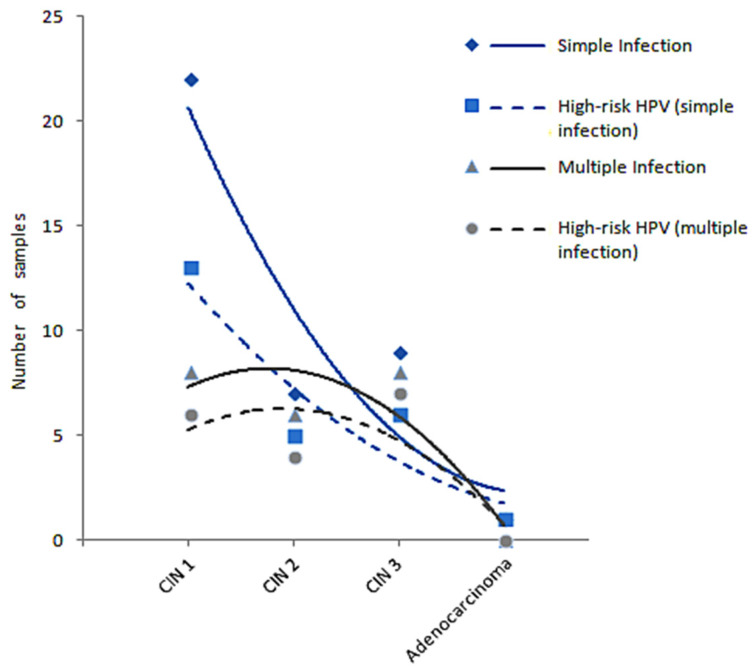
Predominance of hrHPV in single and multiple infections according to disease progress from cervical intraepithelial neoplasia (CIN 1, CIN2, and CIN3) to adenocarcinoma.

**Table 1 microorganisms-12-02326-t001:** Sequences of consensus and degenerate primers used in the outer PCR for the general amplification of all HPV genotypes along with β-globin (BG) PCR primers as a human constitutive gene internal control.

Type	ID	Sequences *	Source	Amplicon Size (bp)
Consensus	MY09	CGT CCM AAR GGA HAC TGA TC	[11]	450
Degenerated	MY09-A	CGT CCM ARR GGA TAC TGA TC	Present Study	450
	MY09-B	CKN CCH ARD GGA AAC TGA TC		
	MY09-C	CKM CCH ARK GGA WTA TGA TC		
	MY09-D	CKD CCY ARD GGR AAT TGG TC		
	MY11-1	GCN GAG GGH CAC AAT AAT GG		
	MY11-2	GCN CAG GGH CAB AAC AAT GG		
	MY11-3	CCH CAR GGH CAT AAT AAT GG		
	MY11-4	GCN CAG GGH CAT AAC AAT GG		
	MY11-5	GCY CAG GGH YWM AAC AAT GG		
Constitutive gene	BG1	CAC CTT TGC CAC ACT GAG TGA G	Present Study	366
	BG2	AGT AAT GTA CTA GGC AGA CTG TG		

Nucleotides degeneration: N = A/C/G/T; D = G/A/T; H = A/T/C, R = A/G, K = G/T, M = A/C, W = A/T, Y = C/T, B = C/G/T. * Primer sequences are detailed in patent application PCT/BR2018/050062 filed on 8 March 2018. WellRed D3-PA (685–706 nm) or HEX (5′-Hexachlorofluorescein phosphoramidite; 535–556 nm) are used for detection.

**Table 2 microorganisms-12-02326-t002:** Sequences of primers used in the single-tube nested-multiplex PCR assay for the amplification of 40 HPV-specific types, with the β-globin (BG) gene serving as a human constitutive gene internal control.

Oncogenic Risk	HPV	Amplicon Size (bp)	Oligonucleotide Sequences (5′ -> 3′) ^1,2^	GenBank Identifier	Nucleotide Position
Low	06	319	(D4-PA) ^3^ ACA TGA CAT TAT GTG CAT CCG	AF092932	6799-6819
			CCT CCC AAA AAC TAA GGT TC		7119-7100
Low	11	290	(D4-PA) ^3^ CAT CTG TGT CTA AAT CTG CT	AF335603.1	1028-1047
			TAT CCT TAT AGG GAT CCT GT		1318-1299
High	16	175	(D4-PA) ^3^ CTA ACT TTA AGG AGT ACC TAC	AY177679	1073-1093
			TCT TCT AGT GTG CCT CCT		1247-1230
High	18	239	(D4-PA) ^3^ TCT CCT GTA CCT GGG CA	U45891	103-119
			GGT AAT AGC AAC AGA TTG TG		340-320
High	26	286	CTG ACA GGT AGT AGC AGA GT	X74472	6877-6858
			(D3-PA) ^3^ ACC ACC CGC AGT ACT AAC CTT		6592-6613
High	30	171	GGT GAC AAT CCA ATA TTC CAG C	X74474	6871-6850
			(D4-PA) ^3^ CGT TAT CCA CAT ATA ATT CAA GCC		6700-6723
High	31	313	(D3-PA) ^3^ CAA TTG CAA ACA GTG ATA CT	U37410	2737-2717
			AAA TTA ACC TCC CAA AAT AC		2424-2444
High	33	315	(D3-PA) ^3^ TTT ATG CAC ACA AGT CAC TAG	U45897-M12732	84-104
			CAC TTC CCA AAA TGT ATA TTT ACC		399-376
High	34	170	(D3-PA) ^3^ ACA ATC CAC AAG TAC AAC TGC	X74476	6542-6562
			TCC ACT GTT CCA ATA TAC TAG AA		6711-6689
High	35	277	(D3-PA) ^3^ TG TTC TGC TGT GTC TWC TAG	M74117-X74477	6609-6628
			GGT TTT GGT GCA CTG GGT		6885-6866
High	39	237	(D2-PA) ^3^ TAC CTC TAT AGA GTC TTC CA	U45905-U45904	90-109
			AGA CTG TAA GTA TCT GTA AGT G		327-306
Low	42	285	(D4-PA) ^3^ TGT GCC ACT GCA ACA TCT GG	M73236	6869-6888
			GGA TCC TTT TTT TCT GGC GTT GT		7152-7130
Low	43	246	CAT GCA ATG GCC TTG TTA GAC	U12504	341-321
			(D4-PA) ^3^ CTG ACC CTA CTG TGC CCA G		98-116
Low	44	205	TAA GGT ACC ATT TGG GGG CG	U12493	300-281
			(D4-PA) CCA CTA CAC AGT CCC CTC CG		95-114
High	45	193	GAA ATC CTG TGC CAA GTA C	X74479-U45915	6662-6680
			(D3-PA) ^3^ TGT AGT AGG TGG TGG AGG G		6855-6837
High	51	337	(D4-PA) ^3^ ACT ATT AGC ACT GCC ACT G	M62877-U45917	6547-6565
			AAT CGT TCC TTT AAA TCA ACA TC		6884-6862
High	52	191	(D4-PA) ^3^ CTG AGG TKA AAA AGG AAA GC	X74481	6694-6715
			ACG GTG GTG GGG TAA GG		6885-6869
High	53	127	(D4-PA) ^3^ TTC CGC AAC CAC ACA GTC	X74482	6807-6790
			TAA CCT CAG CAG ACA GG G		6680-6698
Medium	54	213	GTG TGC TAC AGC ATC CAC GC	U37488	6627-6646
			(D4-PA) ^3^ TCC TCC AAA CTA CTT GTA GC		6839-6820
Medium	55	278	(D4-PA) ^3^ CTT TGC CTT TTC AGG GGG A	U12494	369-351
			GCT GCT ACA ACT CAG TCT CC		91-110
High	56	267	(D4-PA) ^3^ TAG TAC TGC TAC AGA ACA G	X74483	6625-6645
			TTT GGT GGC TGT TCC CG		6892-6878
Low	57	235	(D4-PA)^3^ AAT ACC TGT AGG TGT CCT GC	X55965	6784-6762
			GTC TCT TTG TGT GCC ACT GTA AC		6977-6996
High	58	124	(D3-PA) ^3^ CAC TGA AGT AAC TAA AGA AGA	AY098920	38-58
			CAT TAC CTC TGC AGT TAG TGT		164-144
High	59	213	AAG TTA CAG CAG CAG ATT GA	AF374230	340-321
			(D3-PA) ^3^ CCT ACC AGT TTT AAA GAA TAT GC		127-149
Medium	61	174	(D3-PA) ^3^ ATT TGT ACT GCT ACA TCC CC	U31793	6796-6815
			GAG TCA TCC AAC AAG GCC		6970-6954
Medium	62	249	GAG ACT CGA AAT AGT GAT ATG TC	U12499	322-300
			(D4-PA) ^3^ CTA ATT TTA CTA TTT GTA CCG CC		74-96
Medium	64	110	TCC ACT GTT CTA ATA TAC TAG A	U12495	268-247
			(D4-PA) ^3^ TGC AGA AGA GTA TGA CCT CG		159-178
High	66	283	CCC AAA ACT TAT ATT TAG CCA GG	U01533	374-352
			(D3-PA) ^3^ CTA AAT ATG ATG CCC GTG AAA TC		92-112
High	67	141	CAT AAC ATT TGC AGT AAG GGA	U12492	217-196
			(D4-PA) ^3^ AAC ATG ACT TTA TAT TCT GAG G		76-97
High	68	239	(D3-PA) ^3^ TAC TAC TAC TGA ATC AGC TG	U45934	90-109
			CCG CTA TCT GCA ATC AGT		312-329
Medium	69	237	(D3-PA) ^3^ CTA CCC GCA GTA CCA ACC TC	AB027020-U12497	6550-6570
			AAC TAG CAG TAG GAG GCA AG		6786-6767
Medium	70	209	GCC ATA CCT GCT GTA TAT AG	U22461	1076-1097
			(D3-PA) ^3^ CCT ATA CGT GTC CAC TAAG		1285-1267
Medium	71	202	(D3-PA) ^3^ CTG TGC TAC CAA AAC TGT TGA G	AY330623	6877-6898
			AGC AGT AGG AGG TGG TAA GG		7078-7059
Medium	72	74	(D4-PA) ^3^ ACT ATT TGT ACT GCC ACA GCG	X94164	6819-6839
			GTG TGG CGA AGA TAC TCA CG		6892-6873
High	73	180	GTA GGT ACA CAG GCT AGT AG	AF459425-X94165	71-90
			(D4-PA) ^3^ ATT CCA CTC TTC CAA TAT AGT AG		250-228
Medium	74	329	(D4-PA) ^3^ ACA CGY AGT ACT AAC ATG ACW G	AF436130-U40822	6587-6606
			AAA TTK GCA TAG GGA TTR GGC		6915-6895
Medium	81	140	(D3-PA) ^3^ CTA TTT GCA CAG CTA CAT CTG	EF626590-AJ620209	83-103
			GTA GGC CAT AAT TTC TGG TGT		222-202
High	82	261	(D4-PA) ^3^ CAT TAG CAC TGC TGY TAC TCC	AB027021	6599-6619
			CCY KKT GAC AGG AKG TTG CTG		6859-6844
Medium	83	305	GGA TCC TTT TTA GGG GCA GG	U12489	374-355
			(D4-PA) ^3^ AGT ACC AAT ATT ACT ATT TCA GC		70-92
Medium	84	83	(D4-PA) ^3^ AGG AAT ACC TAA GAC ATG TG	U12490	134-153
			CA TGA CCT CTG GAG TCA G		216-199
β-globin	BG2	366	AGT AAT GTA CTA GGC AGA CTG TG	U01317	62945-62967
	BG3		(D3-PA)-AAG CTG CAC GTG GAT CCT GAG		62601-62622

^1^ Nucleotide degeneration: K = G/T, W = A/T, Y = C/T. ^2^ Primers under the patent application PCT/BR2018/050062, filed on 8 March 2018. ^3^ Fluorophore-labeled probes (Excitation-Emission): WellRed D4-PA (650–670 nm) or FAM: 6-carboxyfluorescein (495–520 nm). WellRed D3-PA (685–706 nm) or HEX: 5′-Hexachlorofluorescein phosphoramidite (535–556 nm). WellRed D2-PA (750–770 nm) or TAMRA: 6-carboxytetramethylrhodamine (557–583 nm).

**Table 3 microorganisms-12-02326-t003:** Comparative results among the HPV diagnostic assays, including the nested-multiplex PCR (NMPCR), Hybrid Capture 2 (HC2), and cytopathological analysis of 40 patients from São Luís, MA, Brazil.

NMPCR × HC2 × Cytopatholgy					
NMPCR	Total	HC2		Cytopathology	
		P	N	P *	N
P	30 (75.0%)	10 (25.0%)	20 (50.0%)	14 (35.0%)	16 (40.0%)
N	10 (25.0%)	2 (5.0%)	8 (20.0%)	2 (5.0%)	8 (20.0%)
Total	40 (100%)	12 (30%)	28 (70.0%)	16 (40.0%)	24 (60.0%)
HC2 × Cytopathology			P	6 (15.0%)	6 (15.0%)
			N	10 (25.0%)	18 (45.0%)
			Total	16 (40.0%)	24 (60.0%)

P indicates positive; N indicates negative. * Positive for cervical intraepithelial neoplasia (CIN).

**Table 4 microorganisms-12-02326-t004:** Prevalence of HPV genotypes * detected by the NMPCR assay in 3079 infected women out of 5223 (59.0%), Brazil.

HPV Genotype/Risk	Single Infection (%)	Multiple Infection (%)
High-Risk (58.6%; 757 out of 1292)	42.0% (1292 out of 3079)	58.0% (1787 out of 3079)
16	4.9	13.4
18	1.6	8.4
26	0.4	3.1
30	2.6	8.1
31	5.2	14.6
33	1.2	5.2
34	1.9	5.7
35	4.0	9.9
39	1.7	12.0
45	1.2	4.5
51	3.2	14.4
52	5.4	18.3
53	7.1	16.1
56	3.7	11.8
66	3.9	12.3
67	0.9	4.8
68	0.5	3.9
69	0.8	4.0
70	1.5	4.3
73	3.7	11.4
82	0.4	1.9
Low- and Medium-Risk (41.4%; 535 out of 1292)		
6	15.1	27.5
11	0.9	3.3
42	4.5	16.1
43	0.3	1.5
44	2.4	8.8
54	4.0	16.5
61	4.6	15.1
62	3.1	10.6
71	1.6	4.6
72	0.1	0.4
74	0.2	0.8
81	1.2	7.9
83	1.2	4.0
84	1.5	3.8

* Five HPV genotypes were observed in very few cases (<0.075%), with genotype 55 in two cases, genotypes 58 and 59 in one case each, and genotypes 57 and 64 in no cases.

## Data Availability

The epidemiological data are not available as they consist of confidential patient test results from BioGenetics Diagnósticos Laboratories. Other original contributions presented in the study are included in the article and Appendix A. Further inquiries can be directed to the corresponding author.

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
