# Peer review of "Expanded HPV Genotyping by Single-Tube Nested-Multiplex PCR May Explain HPV-Related Disease Recurrence"

_microorganisms, 2024, doi:10.3390/microorganisms12112326_

Round 1
Reviewer 1 Report
Comments and Suggestions for Authors
The topic described in this manuscript is innovative, diagnostically relevant and clinically usable. In order to improve the manuscript, it is crucial that the authors visualize the methodology more clearly from the beginning of the execution to the visualization. Some figures are omitted or not visible in this version of the work.
Lines 90-94: Although the authors hypothesized, literature data that exists on this topic should be added here.
Why did the authors choose this manual in-house extraction method over a reliable commercial kit or even automated extraction? It is not clear how many samples this type of extraction was performed on?
Figure 1 is incomplete because it does not visualize the entire method. Since it is new and the detection is based on electrophoresis, a more comprehensive figure is necessary to understand the performance and detection.
Not all parts of the Figure 3 are visible.
Author Response
Dear Reviewer,
Thank you for your comments, please find the point-by-point replies in the attachment.

Reviewer 2 Report
Comments and Suggestions for Authors
Luiz Ricardo Goulart et al. have developed a high-throughput system using a single-tube nested-multiplex polymerase chain reaction (NMPCR) to detect 40 HPV genotypes via capillary electrophoresis. Although the NMPCR test is a promising tool for HPV diagnosis and cervical cancer screening, as demonstrated through a series of simple yet effective experiments, a few critical points need to be addressed.
1) [ Lines 37] The authors mentioned that the assay could identify viral subtypes in 36 multiple infections, even with low viral loads. Mention the lowest detection limit or viral loads the developed assay can detect.
2) [ Lines 157] Indicated if the DNA extraction method was in-house. In the same vein [line 158], 500ul of lysis buffer used, indicate lysis buffer composition or mention catalog # if any commercial buffer was used.
3) [ Lines 187] The Author mentioned 7ul of DNA. Indicate the DNA concentration and the minimum DNA concentration that could be used.
4) Please format Figure 3 to ensure all panels are visible and simplify the legend.
5) The authors should mention the assay's detection limit and, if possible, include the result in the figure.
The authors should provide information about the cross-reactivity of the primers used in the assay with respect to HPV-negative samples. A simple assay, such as testing an HPV-negative sample and confirming the absence of a band, would be appropriate
Author Response

(The authors gave the same response as above.)
